# Elastic Wave Propagation Control in Porous and Finitely Deformed Locally Resonant Nacre-like Metamaterials

**DOI:** 10.3390/ma17030705

**Published:** 2024-02-01

**Authors:** Umberto De Maio, Fabrizio Greco, Paolo Nevone Blasi, Andrea Pranno, Girolamo Sgambitterra

**Affiliations:** Department of Civil Engineering, University of Calabria, 87036 Arcavacata, CS, Italy; umberto.demaio@unical.it (U.D.M.); paolo.nevoneblasi@unical.it (P.N.B.); andrea.pranno@unical.it (A.P.); girolamo.sgambitterra@unical.it (G.S.)

**Keywords:** bioinspired materials, Bloch wave analysis, nonlinear homogenization, band gap, locally resonant, metamaterials, instability, finite deformations

## Abstract

Recent studies have shown that the mechanical properties of bioinspired periodic composite materials can be strongly influenced by finite deformation effects, leading to highly nonlinear static and dynamic behaviors at multiple length scales. For instance, in porous periodic nacre-like microstructures, microscopic and macroscopic instabilities may occur for a given uniaxial loading process and, as a consequence, wave attenuation properties may evolve as a function of the microstructural evolution, designating it as metamaterials. The numerical outcomes provide new opportunities to design bioinspired, soft composite metamaterials characterized by high deformability and enhanced elastic wave attenuation capabilities given by the insertion of voids and lead cores.

## 1. Introduction

Over the last twenty years, as a result of the high mechanical, thermal, and electrical performance requirements [1], advanced composite materials have been preferred over traditional ones in many extreme engineering applications [2,3,4]. For example, structural members or engine parts of space shuttles or aircraft have been designed thanks to advanced composite materials. In addition, recently, bioinspired microstructures were used by researchers to develop innovative materials with exceptional properties. For instance, it has been found that nacre-like materials possess unique mechanical properties resulting from the alternating layering of soft protein and hard aragonite platelets [5,6] and, thanks to the recent development in additive manufacturing [7,8,9], many researchers have started to investigate the mechanical properties of 3D-printed microstructured bioinspired materials [10,11,12,13]. Due to their complex microstructures, such bioinspired composite materials are generally regarded as highly heterogeneous media liable to several nonlinear phenomena such as microscopic and macroscopic instabilities caused by large deformations [14] or, also, microscopic damage processes could occur as a result of platelets debonding from the matrix [15]. Several studies have demonstrated that such microscopic damage phenomena are closely related to fracture phenomena occurring at the macroscopic scale, such as delamination and crack propagation, and that they strongly influence the dynamic response of the structure in terms of natural frequency vibrations [16], thus, representing the most frequent failure precursor for advanced materials employed in extreme engineering applications. Hyperelastic constitutive laws are commonly employed to accurately predict the mechanical behavior of such advanced composite materials in a large deformation framework. To mitigate the need for computationally demanding numerical modeling, various sophisticated numerical modeling strategies have been introduced. These include approaches such as nonlinear homogenization [17] and multiscale methods [18,19]. In recent works, it has been demonstrated that bioinspired nacre-like composite materials may be optimized to improve their mechanical performance under static loadings [20,21], and, due to their periodic microstructures, they are also intrinsically capable of influencing elastic wave propagation. For this reason, their vibrational response is attracting extraordinary attention [22,23,24]. This context has led to the development of innovative bioinspired nacre-like metamaterials. These are composite materials distinguished by their periodic microstructures, which draw inspiration from natural patterns [25,26]. While metamaterial research has made remarkable progress over the past decade, several drawbacks remain. For example, their material characteristics frequently fail to fulfill the energy dissipation criteria essential for managing the transmission of waves and reducing noise in mechanically challenging procedures.

In recent years, the study of elastic wave propagation in nacre-like composite metamaterials has been focused on finding the optimal combination of material and geometric parameters [27,28]. The aim of this study is twofold. Firstly, it seeks to expand the scientific understanding of nacre-like composite metamaterials, especially those that show microscopic instability under extreme loading conditions. Secondly, it introduces a novel, lighter version of a nacre-like composite metamaterial, featuring hollow reinforcing platelets with lead cores. This study investigates how various microscopic materials and geometric parameters, along with the incorporation of lead cores, affect the material’s wave attenuation properties.

Focusing on the control of elastic wave propagation in metamaterials, it is insightful to draw parallels with electromagnetic wave interactions with near-field obstacles that can interact with the propagation of elastic waves in a few ways: (i) the propagating wave can be scattered in different directions, depending on the size, shape and material properties of the obstacle; (ii) if the obstacle material has damping properties, it can absorb some of the wave energy, reducing the wave amplitude; (iii) in certain conditions, the interaction between the wave and the obstacle can lead to resonance, increasing the elastic wave amplitude at specific frequencies. This work definitively parallels the innovative approaches seen in electromagnetic applications, such as the use of hyperbolic dielectric metamaterials to improve focusing capabilities [29] and the intriguing effects of cylindrical scatterers on propagation characteristics within metamaterial slab antennas [30], highlighting the diverse and impactful potential of metamaterials in various scientific domains.

A brief recap of the theoretical concepts related to the nonlinear static and dynamic response of periodic composite materials is reported in Section 2. The numerical results obtained by superimposing an elastic wave motion through the Bloch wave technique [31,32,33] on a finitely deformed configuration of the proposed lightened composite metamaterials are reported in Section 3. As a result of this work, we highlight that there is excellent design potential for porous advanced bioinspired locally resonant metamaterials characterized by excellent wave absorption properties provided by the addition of lead cores and the onset of microscopic instabilities.

## 2. Theoretical Background

In this section, the theoretical background related to the evaluation of the homogenized properties (Section 2.1) is reported together with the problem statement of the Bloch wave analysis employed to investigate the micro- and macroscopic instability conditions and the elastic wave propagation in prestressed periodic materials (Section 2.2).

### 2.1. Homogenized Properties in Periodic Media

Consider a representative volume element (RVE) characterized by a representative unit cell (RUC) of a lightened nacre-like composite material, as reported in Figure 1, consisting of alternating stiff platelets (gray areas), matrix (light gray areas), voids (white areas) and lead cores (dark gray areas).

The volume of the homogenized nacre-like composite material is denoted by ${\bar{V}}_{\left(i\right)}$ and it is enclosed by the surface $\partial {\bar{V}}_{\left(i\right)}$ on which the macroscopic first Piola–Kirchhoff traction vector acts ${\bar{t}}_{R}$. All the quantities reported in Figure 1 with the subscript (*i*) are referring to the undeformed configuration. The volume of the primitive unit cell (RUC) is identified as $V_{\left(i\right)}$. The undeformed and deformed RVE configurations are associated with an infinitesimal neighborhood of a generic point $\bar{X}$ which is related to the homogenized lightened nacre-like composite material subjected to a macroscopic gradient deformation $\bar{F}(\bar{X})$. The position vectors are defined as X and x, with reference to the undeformed and the deformed RVE configuration, respectively. The microscopic gradient deformation tensor is defined as F=∂x/∂X, while its determinant J≡detF represents the Jacobean of the transformation, representing the volume change measure.

For an easier imposition of the essential boundary condition on the RVE, the microscopic equilibrium problem is formulated in terms of the deformation gradient tensor F and the first Piola–Kirchhoff stress tensor $T_{R}$. By assuming a sufficiently small value of the time-like parameter *t* governing the monotonically increasing loading, the quantities in the incremental form can be considered as rate quantities. Therefore, the microconstituents are characterized by the following incrementally linear relation:(1)$${\dot{T}}_{R}=C^{R}\left[\dot{F}\right],$$
where ${\dot{T}}_{R}$, $C^{R}$ and $\dot{F}$ denote the rate of the first Piola–Kirchhoff stress tensor, the fourth-order tangent moduli tensor, and the rate of the deformation gradient tensor, respectively. Here, we consider a finitely deformed composite material composed by nearly incompressible hyperelastic constituents characterized by a strain energy density W(F). The first and the second derivatives of W(F) represent the stress tensor $T_{R}$ and the tangent moduli tensor $C^{R}$, respectively:(2)$$T_{R}=\frac{\partial W}{\partial F},\;C^{R}=\frac{\partial^{2}W}{\partial F^{2}}.$$

By considering a quasi-static loading process and the absence of volume forces, the equation of motion in the undeformed configuration can be written in the following form:(3)$$\mathrm{Div}T_{R}=0.$$

The main averaging relationships are reported in Figure 1; however, readers are referred to [14] for a comprehensive theoretical background and detailed equations on homogenization theory in periodic elastic metamaterials.

### 2.2. Nonlinear Static and Dynamic Response of Periodic Media

The nonlinear static and dynamic responses of a porous, periodic bioinspired microstructure subjected to uniaxial macroscopic compressive loading processes are analyzed (see also [34,35,36] for additional information). In the former case, the microstructural static equilibrium solution path and the accompanying instability analysis are determined. In contrast, in the latter case, the incremental wave motion problem superimposed on a finitely deformed configuration of the microstructure equilibrium path is considered. The interrelations between the microscopic instabilities occurring at a given uniaxial compression and wave propagation phenomena are also analyzed.

Firstly, the static response was investigated by solving the nonlinear boundary value problem formulated on an adequately chosen RVE, adopting the homogenization theory assumptions together with periodic boundary conditions, as reported in Figure 2. The finite deformed configuration was obtained by imposing the following macroscopic gradient deformation tensor:(4)$$\bar{F}(\beta )=(1-\beta )e_{1}⊗e_{1}+{\left(1-\beta \right)}^{-1}e_{2}⊗e_{2}=\left[\begin{array}{cc} \left(1-\beta \right) & 0\\ 0 & {\left(1-\beta \right)}^{-1} \end{array}\right],$$
with $e_{1}$ and $e_{2}$ denoting the unit basis vectors along the direction **X**_1_ and **X**_2_, respectively, and *β* denoting the load parameter. By imposing that ${\bar{F}}_{22}={{\bar{F}}_{11}}^{-1}$, an acceptable approximation of the incompressibility constraint may be established for low percentages of void volume fractions. Subsequently, the onset of primary instabilities with short (microscopic instability) or long (macroscopic instability) wavelength was detected by solving the following frequency domain wave equation through the Floquet–Bloch theorem:(5)$$\mathrm{Div}\left\{C^{R}[\nabla \dot{\kappa }(X)]\right\}+\rho_{\left(i\right)}\omega^{2}\dot{\kappa }(X)=0,$$
where $\rho_{\left(i\right)}$ is the mass density in the undeformed configuration $V_{\left(i\right)}$; $\omega^{2}$ are the roots of the characteristic equation; $C^{R}$ denotes the nominal tangent moduli tensor determined with reference to a unit cell. The variable $\dot{\kappa }(X)$ is defined by the following relation, and it denotes the wave function in a periodic solid based on the Floquet–Bloch theorem:(6)$$\dot{\kappa }(X)=\hat{\kappa }(X)e^{i\kappa^{0}\cdot X},$$
where $\hat{\kappa }(X)$ denotes a wave function that is periodic on the unit cell and $K^{0}$ denotes the Bloch wave vector defining the direction of the wave propagation and the wavelength.

Primary instabilities are detected when the lowest eigenvalue of Equation (5) first vanishes. Secondly, the dynamic response was investigated through a Bloch wave analysis by evaluating the dispersion relations associated with the proposed periodic microstructure. Specifically, the evolution of the dispersion relations for increasing levels of deformation was performed by superimposing the following Bloch–Floquet boundary conditions on the external boundaries of the unit cell on a finitely deformed configuration:(7)$$\left\{\begin{array}{l} {\dot{u}}_{1}{|}_{right}={\dot{u}}_{1}{|}_{left}e^{i(K_{1}^{0}L+K_{2}^{0}H)}\\ {\dot{u}}_{2}{|}_{right}={\dot{u}}_{2}{|}_{left}e^{i(K_{1}^{0}L+K_{2}^{0}H)} \end{array}\right.,\left\{\begin{array}{l} {\dot{u}}_{1}{|}_{upper}={\dot{u}}_{1}{|}_{lower}e^{i(K_{1}^{0}L+K_{2}^{0}H)}\\ {\dot{u}}_{2}{|}_{upper}={\dot{u}}_{2}{|}_{lower}e^{i(K_{1}^{0}L+K_{2}^{0}H)} \end{array}\right.,$$
where $K_{1}^{0}$ and $K_{2}^{0}$ denote the components of the Bloch wave vector $K^{0}$ which are defined as a function of a scalar parameter *k* ranging from 0 to 4 to swap the Bloch wave vector on the external boundaries of the first Brillouin zone defined by the points Γ-X-W-Y. This procedure enables the determination of the bandgap structure of the periodic unit cell, which is represented through dispersion curves. These curves, which correlate the wavelength of propagating waves with their frequency, have been analyzed to observe how the complete bandgaps evolve in response to applied strains. The numerical analyses were conducted using finite element discretization, employing COMSOL Multiphysics v6.1.

## 3. Numerical Simulations for Different Geometric and Material Parameters

We conducted an investigation into the propagation of elastic waves in two scenarios. Initially, we examined the lightened nacre-like microstructures at the undeformed configuration (Case 1). Subsequently, we analyzed the bifurcated state under increasing loading levels in two distinct microstructures: a standard porous nacre-like microstructure with periodically arranged solid and hollow platelets (Case 2), and a lead-enhanced nacre-like microstructure incorporating periodically arranged lead cores (Case 3).

The geometry of the numerical model is reported in Figure 3, the unit cell length is equal to *L* and its height is equal to *H*. Two different thicknesses for the horizontal *b_h_* and vertical *b_v_* matrix interphases were considered. The platelet’s length and height are equal to *L_p_* and *H_p_*, respectively. The parametric analyses were performed by varying the following geometric parameters: the main volume fraction *v_f_*, the hollow platelets volume fraction *v_f_*_(*hp*)_, the platelets aspect ratio *w_p_* = *L_p_/H_p_*, with *L_p_ =* 10 μm and the interface thickness aspect ratio *w_b_* = *b_v_*/*b_h_* with *b_h_* equal to:(8)$$b_{h}=\frac{(-v_{f}w_{b}-v_{f}w_{p}+\sqrt{v_{f}^{2}w_{b}^{2}-2v_{f}^{2}w_{b}w_{p}+v_{f}^{2}w_{p}^{2}+4v_{f}w_{b}w_{p}})L_{p}}{2v_{f}w_{b}w_{p}}.$$

The thickness of the hollow platelets is defined by the following relation:(9)$$b=\frac{H_{p}}{4}+\frac{L_{p}}{4}-\frac{\sqrt{4H_{p}L_{p}v_{f(hp)}+H_{p}^{2}-2H_{p}L_{p}+L_{p}^{2}}}{4}.$$

The main volume fraction $v_{f}=8L_{p}H_{p}/[(2L_{p}+2b_{v})(4H_{p}+4b_{h})]$ is evaluated by considering the volume occupied by both the platelets and voids. The hollow platelets volume fraction *v_f_*_(*hp*)_ defines the percentage of the voids with reference to the hollow platelets (*v_f_*_(*hp*)_ = 1 denotes full void inclusions while *v_f_*_(*hp*)_ = 0 represents full platelets), while *v_f_*_(*v*)_, *v_f_*_(*p*)_ and *v_f_*_(*l*)_ define the volume fraction of the voids, platelets and the lead, respectively.

In the following, we present the relationships between the geometric quantities previously mentioned:(10)$$\left[\begin{array}{l} v_{f(hp)}=\frac{\left(L_{p}-2b)(H_{p}-2b\right)}{L_{p}H_{p}}\;\mathrm{with}\;b\leq \min(\frac{L_{p}}{2},\frac{H_{p}}{2})\\ v_{f(v)}=v_{f(l)}=\frac{1}{4}v_{f}v_{f}^{hp}\\ v_{f(p)}=\frac{1}{2}v_{f}+\left(\frac{1}{2}v_{f}\right)\left(1-v_{f}^{hp}\right) \end{array}\right..$$

The mechanical behavior of the matrix, the platelets and the lead was modeled employing a neo-Hookean hyperelastic constitutive law based on the following strain energy density function:(11)$$W=\frac{1}{2}\mu (\mathrm{tr}(C)-3)-\mu \ln(J)+\frac{1}{2}\lambda \ln{\left(J\right)}^{2},$$
where *μ* is the initial shear modulus; ***C*** is the right elastic Cauchy–Green tensor; J is the Jacobian while *λ* defines the material compressibility, which is taken to be equal to 1000 *μ* to model the incompressible behavior of material phases. The initial shear modulus of the matrix phase is set equal to *μ_m_* = 0.16 MPa corresponding to Young’s modulus equal to 0.5 MPa (typical of a Tango Plus 3D-printed material), and the initial shear modulus of the platelets is *μ_m_* = *kμ_m_*, while their material density and Poisson’s ratio are set equal to each other as $\rho_{\left(p\right)}=\rho_{\left(m\right)}=1.145\;\mathrm{kg}/m^{3}$ and $\nu_{\left(p\right)}=\nu_{\left(m\right)}=0.49$, respectively. The lead cores are characterized by the following material parameters: $\mu_{\left(l\right)}=4929\mathrm{MPa}$, $\nu_{\left(l\right)}=0.42$ and $\rho_{\left(l\right)}=11340\;\mathrm{kg}/m^{3}$.

### 3.1. Case 1: Lightened Nacre-like Metamaterials without Hollow Platelets and Lead Cores at the Undeformed Configuration

Firstly, the bandgap structure was investigated for the limit case with *v_f_*_(*hp*)_ = 1 (corresponding to the case without hollow platelets and lead cores) at the undeformed configuration for increasing values of the main volume fraction *v_f_* (i.e., from 91% to 99%) and by varying the main geometric and material parameters, as reported in Table 1.

The highlighted areas represent the frequency ranges corresponding to complete bandgaps for which the wave propagation is forbidden in any direction of wave propagation. This first set of parametric investigations was conducted focusing on high values of stiffness ratio for a frequency range equal to 0–150 kHz, while lower values of the stiffness ratio will be subsequently investigated.

The results depicted in Figure 4 demonstrate that a joint aspect ratio *w_b_* = 1, which corresponds to equivalent thicknesses in both vertical and horizontal matrix joints, yields enhanced efficacy in attenuating wave propagation when compared with other examined aspect ratios.

Generally speaking, the numerical outcomes highlighted that the complete bandgaps become wider for increasing values of the main volume fraction *v_f_* from 91% to 99%, while for *v_f_ <* 91%, no bandgaps were found for all the investigated parameters. Additionally, our findings indicate that as the platelets aspect ratio *w_p_* increases, the frequency ranges of the complete bandgaps also increase. Simultaneously, these ranges become wider with higher values of the stiffness ratio *k*. Notably, the lowest stiffness ratio examined *k* = 1000 exhibited band gap structures characterized by extremely sparse and narrow bandgaps; consequently, for brevity, these results have not been included in the manuscript.

The numerical outcomes definitively highlighted that the bandgap ranges can be opportunely tuned by choosing the geometric parameters *v_f_* and *w_p_*, considering that the wider bandgap ranges were obtained with *w_b_* = 1, *w_p_* = 4 and the highest investigate value of *k* = 100,000.

### 3.2. Case 2: Lightened Nacre-like Metamaterials with Hollow Platelets and without Lead Cores for Increasing Levels of Deformation (Standard Microstructure)

The previous set of preliminary investigations reported in Section 3.1 focused on high values of stiffness ratio, which leads to the appearance of bandgaps in a high-frequency range (often outside the acoustic range of frequencies), making such composite metamaterials unsuitable for acoustic applications.

In addition, in a previous work by some of the authors [14], it was obtained that with increasing values of the stiffness ratio and platelets aspect ratio, the critical load factors associated with the macroscopic instability notably decrease, leading to a higher risk of catastrophic failures due to the onset of macroscopic instability phenomena. Subsequently, in light of the obtained bandgap structures reported in Figure 4, the investigation was focused on lower values of the stiffness ratio *k* (leading to ranges of frequency belonging to the acoustic range 20 Hz–20 kHz) and also investigated the influence of the applied deformations together with the onset of instabilities at the microscopic scale. A consistent set of parametric investigations was performed concerning the geometric and material parameters reported in Table 2 to identify the best combination giving the widest bandgaps.

After some preliminary investigation, the best combination of geometric and material parameters, giving a wide bandgap in the undeformed configuration, was found: *L_p_* = 10 mm, *k* = 20, *w_b_* = 50, *w_p_* = 4, *v_f_* = 0.8, *v_f_*_(*hp*)_ = 0.8. First of all, the microscopic and the macroscopic instability analyses were performed, and we found that the microscopic instability occurs before the macroscopic one at a load parametric value equal to 0.09789, representing a percentage of deformation along the compression direction (X_1_ direction with reference to Figure 3) equal to about 10%. Subsequently, as can be seen in Figure 5a, by adding a geometric imperfection in the form of the identified critical mode shape, the bifurcated solution was determined, and the bandgap analysis was performed at each loading step to determine the evolution of the dispersion graphs. The proof that the primary instability is characterized by a local instability (wavelength of the critical mode shape comparable with the unit cell size) was given by the local instability check reported in Figure 5b.

In Figure 6, the evolution of the bandgap structure at the undeformed configuration and for increasing levels of deformation after the onset of the microscopic instability is reported. We observed that at the undeformed configuration, a complete wide bandgap is found in the range from 2.6 kHz to 3.2 kHz; meanwhile, as the level of deformation increases, the dispersion graph evolves considerably, showing a thinning of the bandgaps up to the complete extinction of the one identified at the undeformed configuration. However, new and wider complete bandgaps also appear in lower ranges of frequencies, and numerous partial bandgaps along the X_1_ and X_2_ directions of wave propagation appear due to the prestress state inducing the onset of microscopic instability.

The results obtained through the Bloch wave analyses were also validated by the attenuation test performed by means of frequency-domain simulations in which, for a given exciting frequency, the displacement obtained in input (on the left boundary of the model) and in output (on the right boundary of the model) were compared to determine the attenuation intensity, as reported in Figure 7 (periodic boundary conditions were imposed on the top and bottom boundaries).

Figure 7 highlights that the elastic pressure waves propagating with a frequency equal to 2 kHz (outside the bandgap range) can propagate through the proposed microstructured metamaterials. In comparison, elastic wave propagation is completely forbidden with a frequency equal to 3 kHz (inside the complete bandgap range). Further numerical validations, both at the deformed and undeformed configurations, by the transmittance spectra determination were performed on every investigated microstructure, but for brevity, only the first validation is reported in this text.

### 3.3. Case 3: Lightened Nacre-like Metamaterials with Hollow Platelets and Lead Cores for Increasing Levels of Deformation (Lead-Enhanced Microstructure)

In this section, the numerical results related to the evolution of the bandgap structure in a locally resonant metamaterial are reported. The previously investigated nacre-like microstructure with hollow platelets and voids was modified by adding lead cores. In Figure 8, the dispersion relations at the undeformed and deformed configurations are reported to investigate the intricate interplay between structural modifications and the resulting acoustic properties.

At the undeformed configuration, Figure 8a illustrates the presence of wide and complete bandgaps spanning the frequency ranges from 2 to 3 kHz and from 3.4 kHz to 4.2 kHz, together with two thinner bandgaps along the X_2_ direction of elastic wave propagation. The bandgaps found results to be considerably wider than those obtained for a microstructure without the insertion of lead cores (Figure 6a). Introducing lead cores surrounded by a soft matrix, acting as local resonators, strongly enhances the material’s energy absorption. This is especially noteworthy at lower vibration frequencies, where the lead-enhanced microstructure demonstrates superior performance compared to its non-enhanced microstructure, highlighting the potential of the locally resonant metamaterial to attenuate vibrations within this specific frequency band effectively.

Furthermore, as deformation levels increase, the dispersion graph undergoes significant transformations. Notably, in Figure 8b, there is a discernible widening of the first complete bandgap, with the complete extinction of the second one, which is followed by the appearance of thinner complete bandgaps in low- and high-frequency ranges.

In addition, the results highlight the emergence of a new wider bandgap along the X_1_ direction in an incredibly low-frequency range (0.5 kHz to 1 kHz). These phenomena are directly attributed to the prestress state induced by the structural modifications, leading to the onset of microscopic instability. With its lead-enhanced configuration, the proposed locally resonant metamaterial offers a valuable design alternative to control and tune the elastic wave propagation in bioinspired nacre-like metamaterials.

## 4. Conclusions

In this work, the main goal was to design soft metamaterials inspired by nacre, capable of bearing significant deformations before the onset of microscopic instabilities, together with the capability to attenuate the elastic wave propagation in specified frequency ranges beneficial for advanced engineering applications. To achieve this goal, a comprehensive set of parametric analyses was conducted, systematically varying the main geometric and material parameters of a novel microstructure inspired by nacre and modified by the addition of reinforcing hollow platelets and lead cores. The aim was to identify the optimal combination that would yield superior wave attenuation capabilities.

The numerical findings highlighted that an elevated shear stiffness ratio between the reinforcing platelets and the soft matrix contributes to the appearance of wide and complete bandgaps. Interestingly, we found that the promising attenuation properties are not exclusive to scenarios with high contrast between the platelets and the matrix. Instead, they can also be achieved at lower contrast levels by adequately choosing the percentage value of platelets, lead cores, and void inclusion volume fractions.

Additionally, it has been observed that the initiation of microscopic instability involves a microstructural pattern transformation. This transformation, in conjunction with applied deformations, significantly influences the investigated microstructure’s wave propagation properties, giving rise to the formation of both new complete and partial bandgaps.

This study definitively presents novel opportunities for the design of bioinspired soft composite metamaterials. Incorporating lead cores, acting as local resonators, we found a substantial increase in the wave attenuation properties, contributing to the overall versatility and performance of the engineered locally resonant metamaterials. We demonstrated that by the opportune dimensioning of the proposed unit cell and choice of the constituent materials’ mechanical properties, we could design vibration attenuation systems that were effective across a wide range of frequencies. This versatility extends the applicability of our findings beyond just one field, making them relevant in electromagnetic, civil and mechanical engineering. Such adaptability in frequency range handling opens up possibilities for customized solutions in various industrial and technological applications, providing a bridge between different areas of engineering and showcasing the interdisciplinary nature of metamaterials which improves the practical implications of our findings.

As a future perspective of this work, it would be interesting to gradually modify the geometry of the platelets, introducing rounded edges and eventually transitioning them into spherical or elliptical inclusions. This approach aims to retain the essential nacre-like microstructure while exploring how these smoother, more curved geometries could enhance the material’s properties in terms of wave attenuation capabilities. This transition, inspired by the resonant behaviors of spherical inclusions and the mathematical insights into hard inclusions within soft materials discussed in [37,38], is anticipated to provide a balanced approach. It could potentially improve the resonance characteristics and mechanical performance of the proposed bioinspired metamaterial, while still closely adhering to the principles observed in natural nacre structures.

## Figures and Tables

**Figure 1 materials-17-00705-f001:**
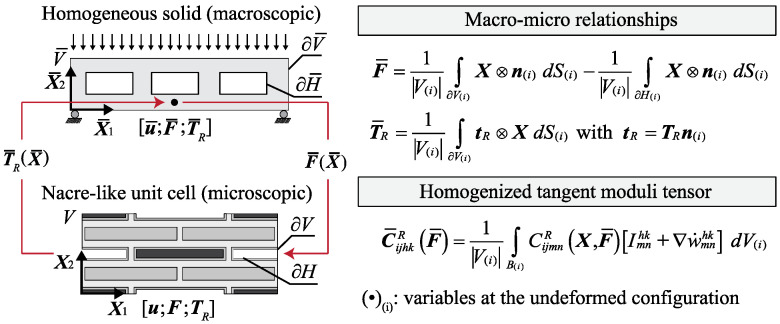
Homogenized solid composite metamaterial, the nacre-like unit cell and the main macro to micro relationships.

**Figure 2 materials-17-00705-f002:**
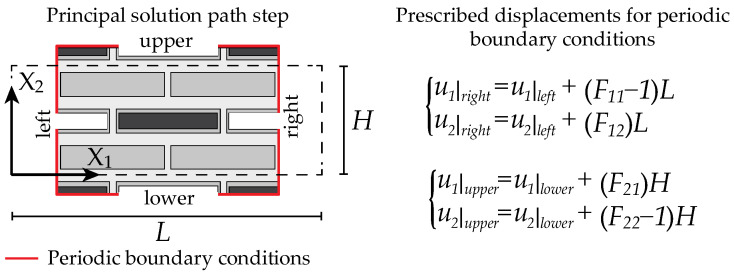
A geometric representation of the RVE adopted in the nonlinear static analysis together with the imposed, prescribed displacement under the assumption of periodic fluctuations.

**Figure 3 materials-17-00705-f003:**
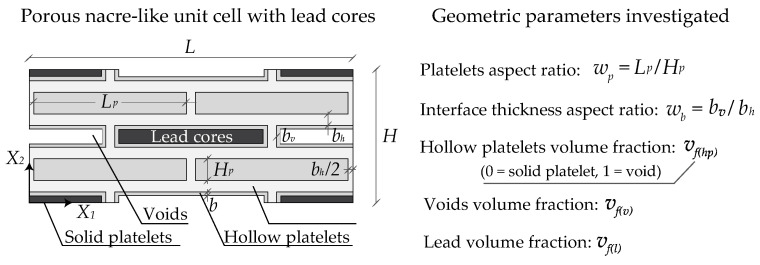
Examined lead-enhanced lightened nacre-like unit cell and the main geometric parameters.

**Figure 4 materials-17-00705-f004:**
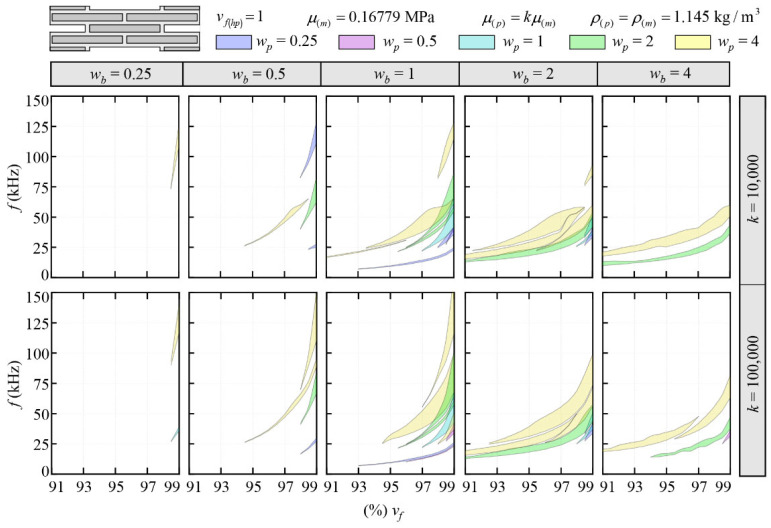
Bandgap structure at the undeformed configuration versus the main volume fraction for different values of stiffness ratio *k*, platelets aspect ratio *w_p_* and joints aspect ratio *w_b_*.

**Figure 5 materials-17-00705-f005:**
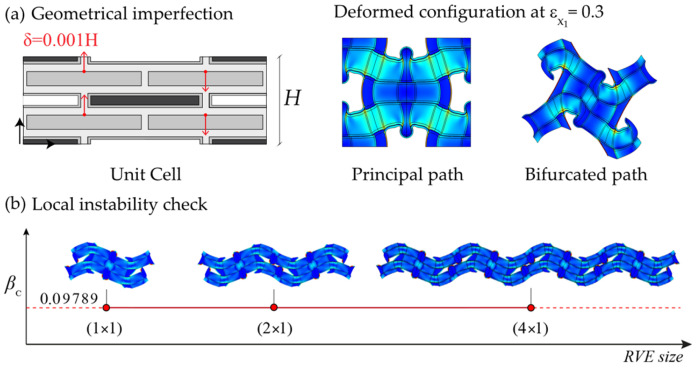
Deformed configuration of the metamaterial RVE for the principal solution path step together with the bifurcated one (**a**) and the results of the local instability check (**b**).

**Figure 6 materials-17-00705-f006:**
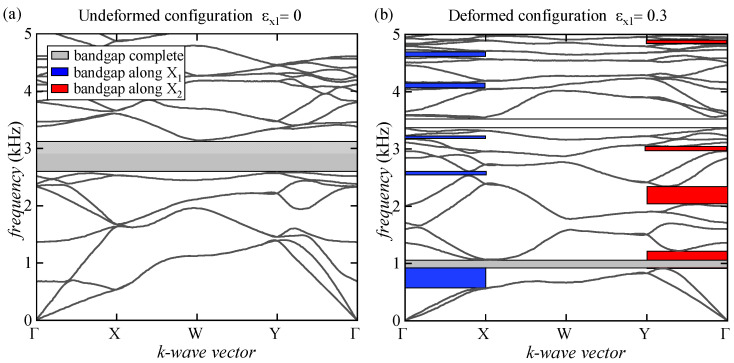
Dispersion relations for the standard microstructure at the undeformed configuration (**a**) and at the bifurcated one (**b**) with a uniaxial stretch ratio along the X_1_ direction equal to 0.3.

**Figure 7 materials-17-00705-f007:**
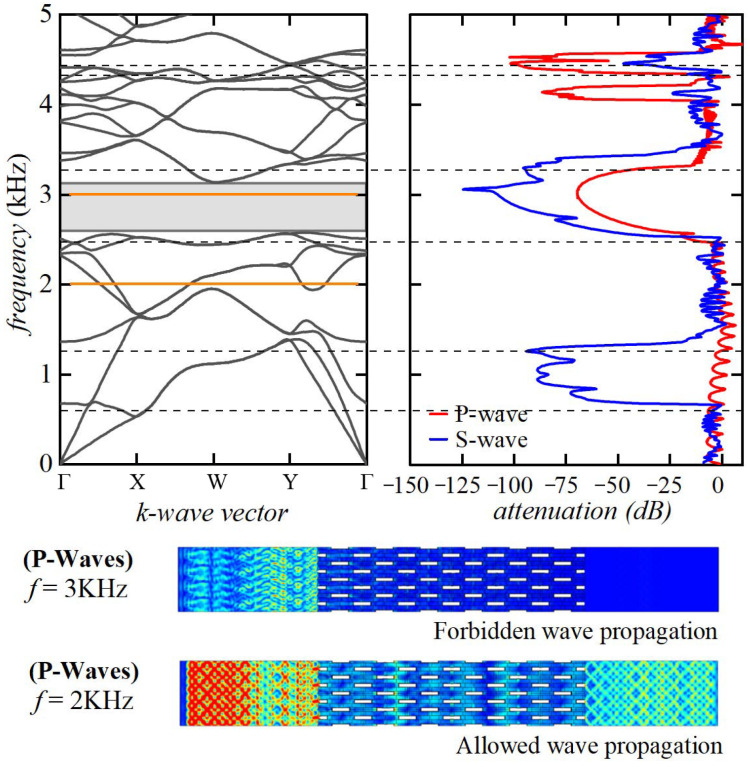
Transmittance spectra at the undeformed configuration and frequency-domain simulation results for propagating waves inside and outside the bandgap range.

**Figure 8 materials-17-00705-f008:**
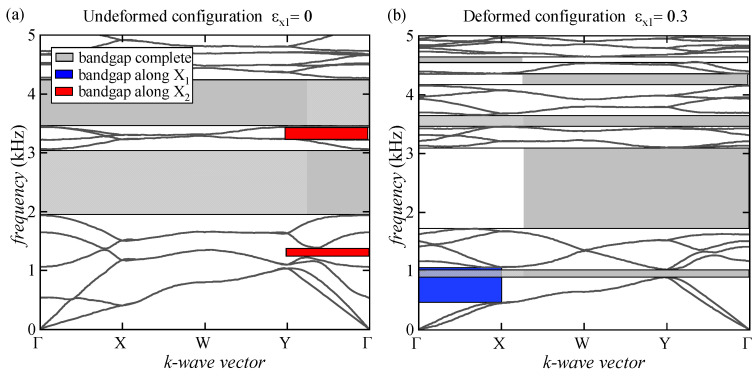
Dispersion relations for the lead-enhanced microstructure at the undeformed configuration (**a**) and at the bifurcated one (**b**) with a uniaxial stretch ratio along the X_1_ direction equal to 0.3.

**Table 1 materials-17-00705-t001:** Examined geometric and material parameters for the composite metamaterial at the undeformed configuration.

** *w_p_* **	0.25	0.5	1	2	4
** *w_b_* **	0.25	0.5	1	2	4
** *k* **	1000	10,000	100,000	-	-

**Table 2 materials-17-00705-t002:** Geometric and material parameters investigated for the standard metamaterial for increasing levels of deformation.

** *k* **	1	5	10	20	30	40	50	100
** *w_b_* **	1	5	10	20	30	40	50	100
** *w_p_* **	4	5	6	7	-	-	-	-
** *v_f_* **	0.5	0.55	0.6	0.65	0.7	0.75	0.8	-
** *v_f(hp)_* **	1	0.95	0.9	0.85	0.8	0.75	0.7	-

## Data Availability

Data are contained within the article.

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
