# Peer review of "Elastic Wave Propagation Control in Porous and Finitely Deformed Locally Resonant Nacre-like Metamaterials"

_materials, 2024, doi:10.3390/ma17030705_

Round 1
Reviewer 1 Report
Comments and Suggestions for Authors
Comments on the Quality of English Languageneed improvement
Author Response
Dear Reviewer 1,
Thank you for your valuable and detailed feedback on our manuscript, titled " Elastic wave propagation control in porous and finitely deformed locally resonant nacre-like metamaterials." We are grateful for the opportunity to enhance our work and appreciate your insights.
Request 1)
In light of your comments about the necessity of an extensive English revision, we have thoroughly reviewed and revised the entire paper, focusing specifically on enhancing the clarity and precision of the English language used.
We understand the importance of clear and professional language in scientific communication and have made every effort to ensure that our manuscript meets this standard. Our revisions include careful attention to grammar, syntax, and technical terminology to improve readability and comprehension.
Despite these efforts, if deemed necessary, we are prepared to seek the assistance of a native English-speaking professional. This additional review will aim to refine the language further, ensuring that our research is presented in the most coherent and polished manner possible to meet the high standards of the “Materials” journal, and we believe that these steps will significantly enhance the quality of our manuscript.
Request 2)
Your suggestion to explore the use of spherical inclusions is indeed significant. We understand that integrating this aspect into our study may significantly improve the behavior of the metamaterials in terms of elastic wave attenuation. We plan to undertake detailed research in this area, recognizing the potential for a broader impact on the field. This exploration of spherical inclusions will be a focus of our future work and will be included in the revised manuscript to discuss future research directions.
Furthermore, we aim to enhance our discussion by referencing the relevant studies you mentioned, specifically the works on bubbles embedded in soft elastic materials (BSM) and hard inclusions embedded in soft elastic materials (HISM). Incorporating these references will provide a more comprehensive theoretical foundation for our research and align our findings with existing literature. We are committed to addressing all the points you have raised and making the necessary revisions to enhance the quality and impact of our work. We believe these changes will significantly contribute to the field and ensure our research aligns with the high standards of your esteemed journal.
We added the following text to the conclusion section: “As a future perspective of this work, it would be interesting to gradually modify the geometry of the platelets, introducing rounded edges and eventually transitioning them into spherical or elliptical inclusions. This approach aims to retain the essential nacre-like microstructure while exploring how these smoother, more curved geometries could enhance the material's properties in terms of wave attenuation capabilities. This transition, inspired by the resonant behaviors of spherical inclusions and the mathematical insights into hard inclusions within soft materials discussed in [32,33], is anticipated to provide a balanced approach. It could potentially improve the resonance characteristics and mechanical performance of the proposed bioinspired metamaterial, while still closely adhering to the principles observed in natural nacre structures.”
Sincerely
Reviewer 2 Report
Comments and Suggestions for Authors
The authors majorly reported the elastic wave attenuation in the presence of different finite deformations in porous and nacre-like metamaterials. Overall, this is quite a well-written manuscript, and I enjoyed reading it smoothly. Sufficient COMSOL simulations were deployed, demonstrating some interesting findings, e.g., the ubiquitous emergence of complete bandgaps regardless of the contrast between the matrix and the platelets.
I only have minor suggestions and recommend its publication in Materials.
1. The homogenization theory of the periodic elastic metamaterials (Sec. 2.1) has a lot of overlap with the one published in Ref. [15]. Figure 1 even looks quite similar to the one in Ref. [15]. It is useful, of course, for the readers to follow the theories from the beginning. However, it might be better to shorten Sec. 2.1 to some degree and to make it more concise. Some of the equations could be omitted by directly citing Ref. [15].
2. The horizontal axis of Fig. 6, i.e., k-vector, could be represented by the solid state physics notations: Γ, X, W, Y points, such that Eq. (19) can be omitted.
3. How were transient simulations as shown in Fig. 7 implemented? Did you launch a sinusoidal signal with a very long duration from the left boundary, and then record the steady-state? Why don't we implement the frequency-domain simulations directly?
4. Some typography errors:
a. The comma at the end of Eq (22) should be a dot. Subsequently, the first letter in Line 144 should then be capitalized.
b. Line 222, "W" in "Where" should be in the lowercase: "where". The dot at the end of Eq (22) should be a comma.
c. Check and unify the indent of the paragraph across the manuscript. For instance, should Line 368 be indented?
Author Response
Dear Reviewer 2,
Thank you for your valuable and detailed feedback on our manuscript, titled " Elastic wave propagation control in porous and finitely deformed locally resonant nacre-like metamaterials." We are grateful for the opportunity to enhance our work and appreciate your insights.
Request 1) The homogenization theory of the periodic elastic metamaterials (Sec. 2.1) has a lot of overlap with the one published in Ref. [15]. Figure 1 even looks quite similar to the one in Ref. [15]. It is useful, of course, for the readers to follow the theories from the beginning. However, it might be better to shorten Sec. 2.1 to some degree and to make it more concise. Some of the equations could be omitted by directly citing Ref. [15].
Response: We appreciate your observation regarding the similarity of Section 2.1 with the content of Ref. [15]. To address your suggestion, we revised this section to make it more concise. This will include a more focused discussion of the essential aspects of the homogenization theory relevant to our work while omitting some of the detailed equations. These omitted details can be directly referred to Ref. [15], thereby reducing redundancy and enhancing the readability of the manuscript. In addition, regarding the similarity of Figure 1 to that in Ref. [15]. We replaced the figure with a new one to better distinguish it from the one in Ref. [15], while still effectively illustrating the concepts of the homogenization theory in periodic elastic metamaterials.
Request 2) The horizontal axis of Fig. 6, i.e., k-vector, could be represented by the solid state physics notations: Γ, X, W, Y points, such that Eq. (19) can be omitted
Response: Representation of k-vector in Fig. 6: Thank you for the suggestion to use solid-state physics notations (Γ, X, W, Y points) for the k-vector in Fig. 6. The figure was replaced with a new one with the right notations and we think that this change not only align with standard notations in the field but also allowed us to omit Eq. (19), thereby streamlining the manuscript.
Request 3): How were transient simulations as shown in Fig. 7 implemented? Did you launch a sinusoidal signal with a very long duration from the left boundary, and then record the steady-state? Why don't we implement the frequency-domain simulations directly?
Response: Thank you for your inquiry regarding the implementation of the simulations in Figure 7. Upon checking the sentence reported in the manuscript, we realized that we had a miscommunication. While we initially mentioned transient simulations, the primary method employed was actually frequency-domain simulations.
Conducting transient simulations for this study would have been computationally demanding and less efficient. Therefore, we opted for frequency-domain simulations, which provided the necessary data more effectively. The transient simulation was conducted just for verification purposes and to generate an animated impact video for potential use in conference presentations.
We appreciate your attention to detail and corrected this error in the manuscript to reflect the simulation methods used in our study accurately.
Typography Errors:
- We corrected the punctuation error at the end of Eq (22) and adjust the subsequent text formatting accordingly.
- The typographical error in Line 222 was corrected by changing "W" to lowercase.
- The manuscript will be thoroughly reviewed for consistency in paragraph indentation. Corrections will be made where inconsistencies like that noted in Line 368 are found.
We are grateful for these constructive comments, as they provide us with an opportunity to enhance the quality and clarity of our manuscript. We believe these revisions will make our work more succinct and reader-friendly, aligning it better with the standards of 'Materials'.
Reviewer 3 Report
Comments and Suggestions for Authors
Deformation effects affect the periodic composite materials that give rise to highly nonlinear static and dynamic behaviors. Wave attenuation properties are tailored in porous periodic microstructures, where various instabilities may occur for a given uniaxial loading process. The homogenized metamaterial and the corresponding undeformed and deformed unit cell configurations are shown while the geometrical representation of the representative volume element adopted for the nonlinear static analysis is also indicated. After numerical simulations, the dispersion relations for the several variants of the investigated microstructures are evaluated.
The paper deals with an interesting topic but certain improving modifications are required in order to become publishable at MDPI Materials. In particular:
(A) More electromagnetic field spatial distributions are required and for a variety of sources. Oblique incidence of plane waves should be considered and demonstration of the bandgaps should be performed.
(B) A larger connection with the potential applications should be made. How the resonances can be exploited towards the tunability of metamaterial deformability and wave attenuation reconfigurability?
(C) The authors should mention how the response of their materials are changing in the presence of near-field obstacles [1,2]. Are the beneficial features still there? If not simulations, an extensive discussion is required.
[1] Free space super focusing using all dielectric hyperbolic metamaterial, Scientific Reports, 2020.
[2] Effect of cylindrical scatterer with arbitrary curvature on the features of a metamaterial slab antenna, Progress in Electromagnetics Research, 2007.
Author Response
Dear Reviewer 3,
Thank you for your valuable and detailed feedback on our manuscript, titled “Elastic wave propagation control in porous and finitely deformed locally resonant nacre-like metamaterials.” We are grateful for the opportunity to enhance our work and appreciate your insights.
We appreciate your suggestions, and we would like to clarify that our study primarily focuses on the attenuation of mechanical elastic waves rather than electromagnetic waves. Therefore, the electromagnetic field spatial distributions may not be directly applicable to our work.
Request (A) More electromagnetic field spatial distributions are required and for a variety of sources. Oblique incidence of plane waves should be considered and demonstration of the bandgaps should be performed.
Our research centers on the attenuation of mechanical elastic waves within porous periodic microstructures enhanced with the addition of lead cores. The bandgap phenomena that we investigate arise due to precompression states and the initiation of local instabilities.
These bandgaps are complete, meaning that they effectively attenuate the propagation of mechanical elastic waves in all directions within the plane of the material. In the manuscript, we provided a comprehensive demonstration and analysis of these complete bandgaps, emphasizing their effectiveness in attenuating mechanical elastic waves and avoiding to explicit the direction of the attenuated elastic waves because this concept is intrinsic in the declaration that the bandgaps are “Complete”. We hope this clarification helps align our research focus with your expectations.
Request (B) A larger connection with the potential applications should be made. How the resonances can be exploited towards the tunability of metamaterial deformability and wave attenuation reconfigurability?
We appreciate your interest in the potential applications of our research. To establish a stronger connection with practical applications, we will expand our discussion on how the resonances identified in our study can be leveraged to enhance the tunability of metamaterial deformability and wave attenuation reconfigurability. Thank you for your insightful feedback. We acknowledge the importance of elucidating the practical applications of our metamaterials, especially in terms of exploiting resonances for tunability and wave attenuation reconfigurability. To address this, we will expand our discussion to clearly illustrate how the unique properties of the porous and finitely deformed nacre-like metamaterials presented in our study can be applied in real-world scenarios.
Specifically, we added a specific comment in the “Conclusion” section to explain the application of these materials in vibration isolation systems, highlighting their potential in industries such as automotive, aerospace, and construction, focusing on the inherent tunability of our metamaterials, stemming from their microstructural design, which can be harnessed to adaptively control vibration and noise levels, which is crucial for enhancing structural integrity and reducing environmental noise pollution. Furthermore, we discussed how the wave attenuation capabilities of our metamaterials can be dynamically modified in response to varying external conditions, thereby offering a versatile solution for managing vibration and noise in a range of practical settings.
These additions will provide a comprehensive understanding of the practical implications of our research, thereby bridging the gap between theoretical findings and potential industrial applications.
Request (C) The authors should mention how the response of their materials is changing in the presence of near-field obstacles [1,2]. Are the beneficial features still there? If not simulations, an extensive discussion is required.
Thank you for your valuable suggestion regarding the response of our metamaterials in the presence of near-field obstacles. Our study primarily focuses on the control of elastic wave propagation in nacre-like metamaterials under mechanical loading, which differs significantly from electromagnetic wave interactions with obstacles. However, we recognize the importance of understanding how similar principles might apply to elastic waves in the presence of physical obstructions.
While our current research does not directly investigate the impact of near-field obstacles on elastic wave propagation, we added a comprehensive discussion on potential implications and similarities that might exist between electromagnetic and mechanical wave interactions with obstacles. This discussion was added in the introduction section, including an overview of existing literature on the subject and a theoretical exploration.
We hope this addition will enhance the scientific character of the manuscript and provide a broader perspective on the potential applications and implications of our research.
Round 2
Reviewer 1 Report
Comments and Suggestions for Authors
I am satisfied with the revised version.
Reviewer 3 Report
Comments and Suggestions for Authors
The authors have updated their paper with new features that render it publishable at Materials.